# ON THE CONVERGENCE OF CERTIFIED ROBUST TRAINING WITH INTERVAL BOUND PROPAGATION

**Yihan Wang\*, Zhouxing Shi\*, Quanquan Gu, Cho-Jui Hsieh**
University of California, Los Angeles
{yihanwang,zshi,qgu,chohsieh}@cs.ucla.edu
*Equal contribution

## ABSTRACT

Interval Bound Propagation (IBP) is so far the base of state-of-the-art methods for training neural networks with certifiable robustness guarantees when potential adversarial perturbations present, while the convergence of IBP training remains unknown in existing literature. In this paper, we present a theoretical analysis on the convergence of IBP training. With an overparameterized assumption, we analyze the convergence of IBP robust training. We show that when using IBP training to train a randomly initialized two-layer ReLU neural network with logistic loss, gradient descent can linearly converge to zero robust training error with a high probability if we have sufficiently small perturbation radius and large network width.

## 1 INTRODUCTION

It has been shown that deep neural networks are vulnerable against adversarial examples (Szegedy et al., 2014; Goodfellow et al., 2015), where a human imperceptible adversarial perturbation can easily alter the prediction by neural networks. This poses concerns to safety-critical applications such as autonomous vehicles, healthcare or finance systems. To combat adversarial examples, many defense mechanisms have been proposed in the past few years (Kurakin et al., 2016; Madry et al., 2018; Zhang et al., 2019; Guo et al., 2018; Song et al., 2018; Xiao et al., 2020). However, due to the lack of reliable measurement on adversarial robustness, many defense methods are later broken by stronger attacks (Carlini & Wagner, 2017; Athalye et al., 2018; Tramer et al., 2020).

There are recently a line of robust training works, known as **certified robust training (certified defense)**, focusing on training neural networks with certified and provable robustness – the network is considered robust on an example if and only if the prediction is provably correct for any perturbation in a predefined set (e.g., a small $\ell_\infty$ ball) (Wang et al., 2018b; Bunel et al., 2018; Zhang et al., 2018; Wang et al., 2018c; Wong & Kolter, 2018; Singh et al., 2018; 2019; Weng et al., 2018; Xu et al., 2020). Certified defense methods provide provable robustness guarantees without referring to any specific attack and thus do not rely on the strength of attack algorithms.

To obtain a neural network with certified robustness, a common practice is to derive a neural network verification method that computes the upper and lower bounds of output neurons given an input region under perturbation, and then train the model by optimizing the loss defined on the worst-case output from verification w.r.t. any possible perturbation. Many methods along this line have been proposed (Wong & Kolter, 2018; Wong et al., 2018; Mirman et al., 2018; Gowal et al., 2018; Raghunathan et al., 2018a; Zhang et al., 2020a). Among these methods, Interval Bound Propagation (IBP) (Mirman et al., 2018; Gowal et al., 2018) is a simple but effective and efficient method so far, which propagates the interval bounds of each neuron through the network to obtain the output bounds of the network. Most of the latest state-of-the-art certified defense works are at least partly based on IBP training (Zhang et al., 2020a; Shi et al., 2021; Lyu et al., 2021; Zhang et al., 2021).

However, the convergence properties of IBP training remained unknown. For standard neural network training (without considering adversarial perturbation, aka natural training), it has been shown that gradient descent for overparameterized networks can provably converge to a global minimizer with random initialization (Li & Liang, 2018; Du et al., 2019b;a; Jacot et al., 2018; Allen-Zhu et al., 2019; Zou et al., 2018). Compared to standard training, IBP-based robust training has a very

different training scheme which requires a different convergence analysis. First, in the robust training problem, input can contain perturbations and the training objective is defined differently from standard training. Second, IBP training essentially optimizes a different network augmented with IBP computation, as illustrated in Zhang et al. (2020a). Third, in IBP training, the activation state of each neuron depends on the certified bounds rather than the values in standard neural network computation, which introduces special perturbation-related terms in our analysis.

In this paper, we conduct a theoretical analysis to study the convergence of IBP training. Following recent convergence analysis on Stochastic Gradient Descent (SGD) for standard training, we consider IBP robust training with gradient flow (gradient descent with infinitesimal step size) for a two-layer overparameterized neural network on a classification task. We summarize our contributions below:

- We provide the first convergence analysis for IBP-based certified robust training. On a two-layer overparameterized ReLU network with logistic loss, with sufficiently small perturbation radius and large network width, gradient flow with IBP has a linear convergence rate, and is guaranteed to converge to zero training error with high probability.

- This result also implies that IBP converges to a state where the certified robust accuracy measured by IBP bounds tightly reflects the true robustness of the network.

- We show additional perturbation-related conditions required to guarantee the convergence of IBP training and identify particular challenges in the convergence analysis for IBP training compared to standard training.

**Notation** We use lowercase letters to denote scalars, and use lower and upper case boldface letters to denote vectors and matrices respectively. $\mathbb{1}(\cdot)$ stand for the indicator function. For a $d$-dimensional vector $\mathbf{x} \in \mathbb{R}^d$, $\|\mathbf{x}\|_p$ is its $\ell_p$-norm. For two sequences $\{a_n\}$ and $\{b_n\}$, $n > 0$, we have $a_n = O(b_n)$ if and only if $\exists C > 0, \exists N > 0, \forall n > N, a_n \le Cb_n$,. And we have $a_n = \Omega(b_n)$ if and only if $\exists C > 0, \exists N > 0, \forall n > N, a_n \ge Cb_n$.

## 2 RELATED WORK

### 2.1 CERTIFIED ROBUST TRAINING

The goal of certified robust training is to maximize the certified robust accuracy of a model evaluated by provable robustness verifiers. Some works added heuristic regularizations during adversarial training to improve certified robustness (Xiao et al., 2019; Balunovic & Vechev, 2020). More effectively, certified defense works typically optimize a certified robust loss which is a certified upper bound of the loss w.r.t. all considered perturbations. Among them, Wong & Kolter (2018); Mirman et al. (2018); Dvijotham et al. (2018); Wong et al. (2018); Wang et al. (2018a) used verification with linear relaxation for nonlinear activations, and Raghunathan et al. (2018b) used semi-definite relaxation. However, IBP (Mirman et al., 2018; Gowal et al., 2018), which computes and propagates interval lower and bounds for each neuron, has been shown as efficient and effective and can even outperform methods using more complicated relaxation (Lee et al., 2021; Jovanović et al., 2021). Most of the effective certified defense methods are at least partly based on IBP. For example, Zhang et al. (2020a) combined IBP with linear relaxation bounds; Lyu et al. (2021) designed a parameterized activation; Zhang et al. (2021) designed a 1-Lipschitz layer with $\ell_\infty$-norm computation before layers using IBP; Shi et al. (2021) accelerated IBP training with shortened training schedules. As most state-of-the-art methods so far contain IBP as an important part, we focus on analyzing the convergence of IBP training in this paper.

On the theoretical analysis for IBP bounds, Baader et al. (2020) analyzed the universal approximation of IBP verification bounds, and Wang et al. (2020) extended the analysis to other activation functions beyond ReLU. However, to the best of our knowledge, there is still no existing work analyzing the convergence of IBP training.

The aforementioned methods for certified robustness target at robustness with deterministic certification. There are also some other works on probabilistic certification such as randomized smoothing (Cohen et al., 2019; Li et al., 2019; Salman et al., 2019) which is out of our scope.

## 2.2 CONVERGENCE OF STANDARD NEURAL NETWORK TRAINING

There have been many works analyzing the convergence of standard neural network training. For randomly initialized two-layer ReLU networks with quadratic loss, Du et al. (2019b) proved that gradient descent can converge to a globally optimum with a large enough network width polynomial in the data size. Ji & Telgarsky (2019) pushed the requirement of network width to a polylogarithmic function. For deep neural networks, Allen-Zhu et al. (2019) proved that for deep ReLU networks, gradient descent has a linear convergence rate for various loss functions with width polynomial in network depth and data size. Chen et al. (2019) proved that a polylogarithmic width is also sufficient for deep neural networks to converge. However, they only focus on standard training and cannot be directly adapted to the robust training settings.

## 2.3 CONVERGENCE OF EMPIRICAL ADVERSARIAL TRAINING

Robust training is essentially a min-max optimization. For a training data distribution $\mathcal{X}$, the objective for learning a model $f_\theta$ parameterized by $\theta$ can be written as[1]:

$$\arg\min_\theta \mathbb{E}_{(\mathbf{x},y)\sim\mathcal{X}} \max_{\Delta\in\mathbb{S}} \ell(f_\theta(\mathbf{x}+\Delta), y),$$

where $(\mathbf{x}, y)$ is a sample, $\ell(\cdot, y)$ is the loss function, $\mathbb{S}$ is the space of perturbations. Empirical adversarial training approximates the inner minimization by adversarial attacks, and some works analyzed the convergence of adversarial training: Wang et al. (2019) considered a first-order stationary condition for the inner maximization problem; Gao et al. (2019); Zhang et al. (2020b) showed that overparameterized networks with projected gradient descent can converge to a state with robust loss close to 0 and the the inner maximization by adversarial attack is nearly optimal; and Zou et al. (2021) showed that adversarial training provably learns robust halfspaces in the presence of noise.

However, there is a significant difference between empirical adversarial training and certified robust training such as IBP. Adversarial training involves a concrete perturbation $\Delta$, which is an approximate solution for the inner maximization and could lead to a concrete adversarial input $\mathbf{x} + \Delta$. However, in IBP-based training, the inner maximization is computed from certified bounds, where for each layer, the certified bounds of each neuron are computed independently, and thereby the certified bounds of the network generally do not correspond to any specific $\Delta$. Due to this significant difference, prior theoretical analysis on adversarial training, which requires a concrete $\Delta$ for inner maximization, is not applicable to IBP.

## 3 PRELIMINARIES

### 3.1 NEURAL NETWORKS

Following Du et al. (2019b), we consider a similar two-layer ReLU network. Unlike Du et al. (2019b) which considered a regression task with the square loss, we consider a classification task where IBP is usually used, and we consider binary classification for simplicity. On a training dataset $\{(\mathbf{x}_i, y_i)\}_{i=1}^n$, for every $i \in [n]$, $(\mathbf{x}_i, y_i)$ is a training example with $d$-dimensional input $\mathbf{x}_i(\mathbf{x}_i \in \mathbb{R}^d)$ and label $y_i(y_i \in \{\pm 1\})$, and the network output is:

$$f(\mathbf{W}, \mathbf{a}, \mathbf{x}_i) = \frac{1}{\sqrt{m}} \sum_{r=1}^m a_r \sigma(\mathbf{w}_r^\top \mathbf{x}_i), \tag{1}$$

where $m$ is the width of hidden layer (the first layer) in the network, $\mathbf{W} \in \mathbb{R}^{m \times d}$ is the weight matrix of the hidden layer, $\mathbf{w}_r(r \in [m])$ is the $r$-th row of $\mathbf{W}$, $\mathbf{a} \in \mathbb{R}^m$ is the weight vector of the second layer (output layer) with elements $a_1, \cdots, a_m$, and $\sigma(\cdot)$ is the activation function. We assume the activation is ReLU as IBP is typically used with ReLU. For initialization, we set $a_r \sim \text{unif}[\{1, -1\}]$ and $\mathbf{w}_r \sim \mathbf{N}(0, \mathbf{I})$. Only the first layer is trained after initialization. Since we consider binary classification, we use a logistic loss. For training example $(\mathbf{x}_i, y_i)$, we define $u_i(\mathbf{W}, \mathbf{a}, \mathbf{x}_i) := y_i f(\mathbf{W}, \mathbf{a}, \mathbf{x}_i)$,

---

[1]Here we use notations to denote the general robust training problem, but in our later analysis, we will have different notations for a simplified problem setting.

the loss on this example is computed as $l(u_i(\mathbf{W}, \mathbf{a}, \mathbf{x}_i)) = \log(1 + \exp(-u_i(\mathbf{W}, \mathbf{a}, \mathbf{x}_i)))$, and the standard training loss on the whole training set is

$$L = \sum_{i=1}^{n} l(u_i(\mathbf{W}, \mathbf{a}, \mathbf{x}_i)) = \sum_{i=1}^{n} \log\left(1 + \exp(-u_i(\mathbf{W}, \mathbf{a}, \mathbf{x}_i))\right).$$

## 3.2 CERTIFIED ROBUST TRAINING

In the robust training setting, for original input $\mathbf{x}_i$ ($\forall i \in [n]$), we consider that the actual input may be perturbed into $\mathbf{x}_i + \Delta_i$ by perturbation $\Delta_i$. For a widely adopted setting, we consider $\ell_\infty$ perturbations, where $\Delta_i$ is bounded by an $\ell_\infty$ ball with radius $\epsilon (0 \leq \epsilon \leq 1)$, i.e., $\|\Delta_i\|_\infty \leq \epsilon$. For the convenience of subsequent analysis and without loss of generality, we make the following assumption on each $\mathbf{x}_i$, which can be easily satisfied by normalizing the training data:

**Assumption 1.** $\forall i \in [n]$, *we assume there exists some $\xi > 0$, such that $\mathbf{x}_i \in [\epsilon, 1]^d$, $\|\mathbf{x}_i\|_2 \geq \xi$.*

In Du et al. (2019b), they also assume there are no parallel data points, and in our case we assume this holds under any possible perturbation, formulated as:

**Assumption 2.** *For perturbation radius $\epsilon$, we assume that*

$$\forall i, j \in [n], i \neq j, \forall \mathbf{x}_i' \in B_\infty(\mathbf{x}_i, \epsilon), \forall \mathbf{x}_j' \in B_\infty(\mathbf{x}_j, \epsilon), \quad \mathbf{x}_i' \nparallel \mathbf{x}_j',$$

*where $B_\infty(\mathbf{x}_i, \epsilon)$ stands for the $\ell_\infty$-ball with radius $\epsilon$ centered at $\mathbf{x}_i$.*

IBP training computes and optimizes a robust loss $\overline{L}$, which is an upper bound of the standard loss for any possible perturbation $\Delta_i$ ($\forall i \in [n]$):

$$\overline{L} \geq \sum_{i=1}^{n} \max_{\Delta_i} \left\{ \log\left(1 + \exp(-y_i f(\mathbf{W}, \mathbf{a}, \mathbf{x}_i + \Delta_i))\right) \mid \|\Delta_i\|_\infty \leq \epsilon \right\}.$$

To compute $\overline{L}$, since $\log(\cdot)$ and $\exp(\cdot)$ are both monotonic, for every $i \in [n]$, IBP first computes the lower bound of $u_i(\mathbf{W}, \mathbf{a}, \mathbf{x}_i + \Delta_i)$ for $\|\Delta_i\|_\infty \leq \epsilon$, denoted as $\underline{u}_i$. Then the IBP robust loss is:

$$\overline{L} = \sum_{i=1}^{n} \log(1 + \exp(-\underline{u}_i)), \quad \text{where } \underline{u}_i \leq \min_{\Delta_i} u_i(\mathbf{W}, \mathbf{a}, \mathbf{x}_i + \Delta_i) \ (i \in [n]). \tag{2}$$

IBP computes and propagates an interval lower and upper bound for each neuron in the network, and then $\underline{u}_i$ is equivalent to the lower bound of the final output neuron. Initially, the interval bound of the input is $[\mathbf{x}_i - \epsilon \cdot \mathbf{1}, \mathbf{x} + \epsilon \cdot \mathbf{1}]$ given $\|\Delta_i\|_\infty \leq \epsilon$, since $\mathbf{x}_i - \epsilon \cdot \mathbf{1} \leq \mathbf{x}_i + \Delta_i \leq \mathbf{x}_i + \epsilon \cdot \mathbf{1}$ element-wisely holds. Then this interval bound is propagated to the first hidden layer, and we have the interval bound for each neuron in the first layer:

$$\forall r \in [m], \ \sigma\left(\mathbf{w}_r^\top \mathbf{x}_i - \epsilon \|\mathbf{w}_r\|_1\right) \leq \sigma\left(\mathbf{w}_r^\top (\mathbf{x}_i + \Delta_i)\right) \leq \sigma\left(\mathbf{w}_r^\top \mathbf{x}_i + \epsilon \|\mathbf{w}_r\|_1\right).$$

These bounds are further propagated to the second layer. We focus on the lower bound of $u_i$, which can be computed from the bounds of the first layer by considering the sign of multiplier $y_i a_r$:

$$u_i(\mathbf{W}, \mathbf{a}, \mathbf{x}_i + \Delta_i) = y_i \frac{1}{\sqrt{m}} \sum_{r=1}^{m} a_r \sigma(\mathbf{w}_r^\top (\mathbf{x}_i + \Delta_i))$$

$$\geq \frac{1}{\sqrt{m}} \sum_{r=1}^{m} \left\{ \mathbb{1}(y_i a_r = 1) \sigma\left(\mathbf{w}_r^\top \mathbf{x}_i - \epsilon \|\mathbf{w}_r\|_1\right) \right.$$

$$\left. + \mathbb{1}(y_i a_r = -1) \sigma\left(\mathbf{w}_r^\top \mathbf{x}_i + \epsilon \|\mathbf{w}_r\|_1\right) \right\} := \underline{u}_i. \tag{3}$$

Then the IBP robust loss can be obtained as Eq. (2). And we define $\underline{\mathbf{u}} := (\underline{u}_1, \underline{u}_2, \cdots, \underline{u}_n)$.

We define **certified robust accuracy** in IBP training as the percentage of examples that IBP bounds can successfully certify that the prediction is correct for any concerned perturbation. An example $i (i \in [n])$ is considered as robustly classified under IBP verification if and only if $\underline{u}_i > 0$. Let $\tilde{u}_i$ be the *exact* solution of the minimization in Eq. (2) rather than relaxed IBP bounds, we also define the **true robust accuracy**, where the robustness requires $\tilde{u}_i > 0$. The certified robust accuracy by IBP is a provable lower bound of the true robust accuracy.

### 3.3 GRADIENT FLOW

Gradient flow is gradient descent with infinitesimal step size for a continuous time analysis, and it is adopted in prior works analyzing standard training (Arora et al., 2018; Du et al., 2019a;b). In IBP training, gradient flow is defined as:

$$\forall r \in [m], \quad \frac{d\mathbf{w}_r(t)}{dt} = -\frac{\partial \overline{L}(t)}{\partial \mathbf{w}_r(t)}, \tag{4}$$

where $\mathbf{w}_1(t), \mathbf{w}_2(t), \cdots, \mathbf{w}_m(t)$ are rows of the weight matrix at time $t$, and $\overline{L}(t)$ is the IBP robust loss defined as Eq. (2) using weights at time $t$.

### 3.4 GRAM MATRIX

Under the gradient flow setting as Eq. (4), for all $i \in [n]$, we analyze the dynamics of $\underline{u}_i$ during IBP training, and we use $\underline{u}_i(t)$ to denote its value at time $t$:

$$\frac{d}{dt}\underline{u}_i(t) = \sum_{r=1}^{m} \left\langle \frac{\partial \underline{u}_i(t)}{\partial \mathbf{w}_r(t)}, \frac{d\mathbf{w}_r(t)}{dt} \right\rangle = \sum_{j=1}^{n} -l'(\underline{u}_j)\mathbf{H}_{ij}(t), \tag{5}$$

where $l'(\underline{u}_j)$ is the derivative of the loss, $\mathbf{H}(t)$ is a Gram matrix and defined as $\mathbf{H}_{ij}(t) = \sum_{r=1}^{m} \left\langle \frac{\partial \underline{u}_i(t)}{\partial \mathbf{w}_r(t)}, \frac{\partial \underline{u}_j(t)}{\partial \mathbf{w}_r(t)} \right\rangle$ ($\forall 1 \leq i, j \leq n$). We provide a detailed derivation in Appendix B.1. The dynamic of $\underline{u}_i$ can be described using $\mathbf{H}$.

From Eq. (3), $\forall i \in [n], r \in [m]$, derivative $\frac{\partial \underline{u}_i(t)}{\partial \mathbf{w}_r(t)}$ can be computed as follows:

$$\frac{\partial \underline{u}_i(t)}{\partial \mathbf{w}_r(t)} = \frac{1}{\sqrt{m}} y_i a_r \left( A_{ri}^+(t)\left(\mathbf{x}_i - \epsilon \operatorname{sign}(\mathbf{w}_r(t))\right) + A_{ri}^-(t)\left(\mathbf{x}_i + \epsilon \operatorname{sign}(\mathbf{w}_r(t))\right) \right),$$

where $\operatorname{sign}(\mathbf{w}_r(t))$ is element-wise for $\mathbf{w}_r(t)$, and we define *indicators*

$$A_{ri}^+(t) := \mathbb{1}(y_i a_r = 1, \mathbf{w}_r(t)^\top \mathbf{x}_i - \epsilon\|\mathbf{w}_r(t)\|_1 > 0),$$
$$A_{ri}^-(t) := \mathbb{1}(y_i a_r = -1, \mathbf{w}_r(t)^\top \mathbf{x}_i + \epsilon\|\mathbf{w}_r(t)\|_1 > 0).$$

Then elements in $\mathbf{H}$ can be written as:

$$\mathbf{H}_{ij}(t) = \frac{1}{m} y_i y_j \sum_{r=1}^{m} a_r^2 \left( A_{ri}^+(t)\left(\mathbf{x}_i - \epsilon \operatorname{sign}(\mathbf{w}_r(t))\right) + A_{ri}^-(t)\left(\mathbf{x}_i + \epsilon \operatorname{sign}(\mathbf{w}_r(t))\right) \right)^\top$$

$$\left( A_{rj}^+(t)\left(\mathbf{x}_j - \epsilon \operatorname{sign}(\mathbf{w}_r(t))\right) + A_{rj}^-(t)\left(\mathbf{x}_j + \epsilon \operatorname{sign}(\mathbf{w}_r(t))\right) \right)$$

$$= \frac{1}{m} y_i y_j \left( \mathbf{x}_i^\top \mathbf{x}_j \sum_{r=1}^{m} \alpha_{rij}(t) - \epsilon\left( \sum_{r=1}^{m}(\beta_{rij}(t)\mathbf{x}_i + \beta_{rji}(t)\mathbf{x}_j)^\top \operatorname{sign}(\mathbf{w}_r(t)) \right) + \epsilon^2 d \sum_{r=1}^{m} \gamma_{rij}(t) \right), \tag{6}$$

where $\alpha_{rij}(t), \beta_{rij}(t), \gamma_{rij}(t)$ are defined as follows

$$\alpha_{rij}(t) = (A_{ri}^+(t) + A_{ri}^-(t))(A_{rj}^+(t) + A_{rj}^-(t)),$$
$$\beta_{rij}(t) = (A_{ri}^+(t) + A_{ri}^-(t))(A_{rj}^+(t) - A_{rj}^-(t)),$$
$$\gamma_{rij}(t) = (A_{ri}^+(t) - A_{ri}^-(t))(A_{rj}^+(t) - A_{rj}^-(t)).$$

Further, we define $\mathbf{H}^\infty$ which is the elementwise expectation of $\mathbf{H}(0)$, to characterize $\mathbf{H}(0)$ on the random initialization basis:

$$\forall 1 \leq i, j \leq n, \quad \mathbf{H}_{ij}^\infty := \mathbb{E}_{\forall 1 \leq r \leq m, \mathbf{w}_r \sim \mathbf{N}(0,\mathbf{I}), a_r \sim \operatorname{unif}[\{-1,1\}]} \mathbf{H}_{ij}(0),$$

where $\mathbf{H}_{ij}(0)$ depends on the initialization of weights $\mathbf{w}_r$ and $a_r$. We also define $\lambda_0 := \lambda_{\min}(\mathbf{H}^\infty)$ as the least eigenvalue of $\mathbf{H}^\infty$. We will prove that $\mathbf{H}(0)$ is positive definite with high probability, by showing that $\mathbf{H}^\infty$ is positive definite and bounding the difference between $\mathbf{H}(0)$ and $\mathbf{H}^\infty$.

# 4 CONVERGENCE ANALYSIS FOR IBP TRAINING

We present the following main theorem which shows the convergence of IBP training under certain conditions on perturbation radius and network width:

**Theorem 1** (Convergence of IBP Training). *Suppose Assumptions 1 and 2 hold for the training data, and the $\ell_\infty$ perturbation radius satisfies $\epsilon \leq O\left(\min\left(\frac{\delta^2\lambda_0^2}{d^{2.5}n^3}, \frac{\sqrt{2d}R}{\log(\sqrt{\frac{2\pi d}{R}}\xi)}\right)\right)$, where $R = \frac{c\delta\lambda_0}{d^{1.5}n^2}$, $c = \frac{\sqrt{2\pi}\xi}{384}$. For a two-layer ReLU network (Eq. (1)), suppose its width for the first hidden layer satisfies $m \geq \Omega\left(\left(\frac{d^{1.5}n^4\delta\lambda_0}{\delta^2\lambda_0^2 - \epsilon d^{2.5}n^4}\right)^2\right)$, and the network is randomly initialized as $a_r \sim unif[\{1, -1\}], \mathbf{w}_r \sim \mathbf{N}(0, \mathbf{I})$, with the second layer fixed during training. Then for any confidence $\delta(0 < \delta < 1)$, with probability at least $1 - \delta$, IBP training with gradient flow can converge to zero training error.*

The theorem contains two findings: First, for a given $\epsilon$, as long as it satisfies the upper bound on $\epsilon$ specified in the theorem, with a sufficiently large width $m$, convergence of IBP training is guaranteed with high probability; Second, when $\epsilon$ is larger than the upper bound, IBP training is not guaranteed to converge under our analysis even with arbitrarily large $m$, which is essentially different from analysis on standard training and implies a possible limitation of IBP training.

In the following part of this section, we provide the proof sketch for the main theorem.

## 4.1 STABILITY OF THE GRAM MATRIX DURING IBP TRAINING

We first analyze the stability of $\mathbf{H}$ during training since $\mathbf{H}$ can characterize the dynamic of the training as defined in Eq. (5). We show that when there exists some $R$ such that the change of $\mathbf{w}_r(\forall r \in [m])$ is restricted to $\|\mathbf{w}_r(t) - \mathbf{w}_r(0)\|_2 \leq R$ during training, we can guarantee that $\lambda_{\min}(\mathbf{H}(t))$ remains positive with high probability. This property will be later used to reach the conclusion on the convergence. We defer the derivation for the constraint on $R$ to a later part.

For all $r \in [m]$, with the aforementioned constraint on $\mathbf{w}_r(t)$, we first show that during the IBP training, most of $\alpha_{rij}(t), \beta_{rij}(t), \gamma_{rij}(t)$ terms in Eq. (6) remain the same as their initialized values ($t = 0$). This is because for for each of $\alpha_{rij}(t), \beta_{rij}(t), \gamma_{rij}(t)$, the probability that its value changes during training can be upper bounded by a polynomial in $R$, and thereby the probability can be made sufficiently small for a sufficiently small $R$, as the following lemma shows:

**Lemma 1.** $\forall r \in [m]$, at some time $t > 0$, suppose $\|\mathbf{w}_r(t) - \mathbf{w}_r(0)\|_2 \leq R$ holds for some $R$, and $\epsilon \leq \frac{\sqrt{2d}R}{\log(\sqrt{\frac{2\pi d}{R}}\xi)}$ holds, then for all $1 \leq i, j \leq n$, we have

$$\Pr(\alpha_{rij}(t) \neq \alpha_{rij}(0)), \Pr(\beta_{rij}(t) \neq \beta_{rij}(0)), \Pr(\gamma_{rij}(t) \neq \gamma_{rij}(0)) \leq \frac{12}{\sqrt{2\pi}\xi}(1 + \epsilon)\sqrt{d}R \coloneqq \tilde{R}.$$

We provide the full proof in Appendix A.1. Probabilities in Lemma 1 can be bounded as long as the probability that each of indicator $A_{ri}^+(t), A_{ri}^-(t), A_{rj}^+(t), A_{rj}^-(t)$ changes is upper bounded respectively. When the change of $\mathbf{w}_r(t)$ is bounded, the indicators can change during the training only if at initialization $|\mathbf{w}_r(0)^\top\mathbf{x}_i \pm \epsilon\|\mathbf{w}_r(0)\|_1|$ is sufficiently small, whose probability can be upper bounded (notation $\pm$ here means the analysis is consistent for both $+$ and $-$ cases). To bound this probability, while Du et al. (2019b) simply used the anti-concentration of standard Gaussian distribution in their standard training setting, here our analysis is different due to additional perturbation-related terms $\epsilon\|\mathbf{w}_r(0)\|_1$, and we combine anti-concentration and the tail bound of standard Gaussian in our proof.

We can then bound the change of the Gram matrix, i.e., $\|\mathbf{H}(t) - \mathbf{H}(0)\|_2$:

**Lemma 2.** $\forall r \in [m]$, at some time $t > 0$, suppose $\|\mathbf{w}_r(t) - \mathbf{w}_r(0)\|_2 \leq R$ holds for some constant $R$, for any confidence $\delta(0 < \delta < 1)$, with probability at least $1 - \delta$, it holds that

$$\|\mathbf{H}(t) - \mathbf{H}(0)\|_2 \leq \frac{12(1 + \epsilon)(1 + 2\epsilon + \epsilon^2)d^{1.5}n^2}{\sqrt{2\pi}\xi\delta}R. \tag{7}$$

This can be proved by first upper bounding $\mathbb{E}[|\mathbf{H}_{ij}(t) - \mathbf{H}_{ij}(0)|]$ ($\forall 1 \leq i, j \leq n$) using Lemma 1, and then by Markov's inequality, we can upper bound $\|\mathbf{H}(t) - \mathbf{H}(0)\|_2$ with high probability.

We provide the proof in Appendix A.2. And by triangle inequality, we can also lower bound $\lambda_{\min}(\mathbf{H}(t))$:

**Corollary 1.** $\forall r \in [m]$, *at some time* $t > 0$, *suppose* $\|\mathbf{w}_r(t) - \mathbf{w}_r(0)\|_2 \leq R$ *holds for some constant* $R$, *for any confidence* $\delta(0 < \delta < 1)$, *with probability at least* $1 - \delta$, *it holds that*

$$\lambda_{\min}(\mathbf{H}(t)) \geq \lambda_{\min}(\mathbf{H}(0)) - \frac{12(1 + \epsilon)(1 + 2\epsilon + \epsilon^2)d^{1.5}n^2}{\sqrt{2\pi}\xi\delta}R, \tag{8}$$

*where* $\lambda_{min}(\cdot)$ *stands for the minimum eigenvalue.*

We also need to lower bound $\lambda_{\min}(\mathbf{H}(0))$ in order to lower bound $\lambda_{\min}(\mathbf{H}(t))$. Given Assumption 2, we show that the minimum eigenvalue of $\mathbf{H}^\infty$ is positive:

**Lemma 3.** *When the dataset satisfies Assumption 2,* $\lambda_0 := \lambda_{\min}(\mathbf{H}^\infty) > 0$ *holds true.*

The lemma can be similarly proved as Theorem 3.1 in Du et al. (2019b), but we have a different Assumption 2 considering perturbations. We discuss in more detail in Appendix A.3. Then we can lower bound $\lambda_{\min}(\mathbf{H}(0))$ by Lemma 3.1 from Du et al. (2019b):

**Lemma 4** (Lemma 3.1 from Du et al. (2019b)). *If* $\lambda_0 > 0$, *for any confidence* $\delta(0 < \delta < 1)$, *take* $m = \Omega(\frac{n^2}{\lambda_0^2}\log(\frac{n}{\delta}))$, *then with probability at least* $1 - \delta$, *it holds true that* $\lambda_{\min}(\mathbf{H}(0)) \geq \frac{3}{4}\lambda_0$.

Although we have different values in $\mathbf{H}(0)$ for IBP training, we can still adopt their original lemma because their proof by Hoeffding's inequality is general regardless of values in $\mathbf{H}(0)$. We then plug in $\lambda_{\min}(\mathbf{H}(0)) \geq \frac{3}{4}\lambda_0$ to Eq. (8), and we solve the inequality to find a proper $R$ such that $\lambda_{\min}(\mathbf{H})(t) \geq \frac{\lambda_0}{2}$, as shown in the following lemma (proved in Appendix A.4):

**Lemma 5.** *For any confidence* $\delta(0 < \delta < 1)$, $\forall r \in [m]$, *suppose* $\|\mathbf{w}_r(t) - \mathbf{w}_r(0)\|_2 \leq R$ *holds, where* $R = \frac{c\delta\lambda_0}{d^{1.5}n^2}$ *with* $c = \frac{\sqrt{2\pi}\xi}{384}$, *then probability at least* $1 - \delta$, $\lambda_{\min}(\mathbf{H}(t)) \geq \frac{\lambda_0}{2}$ *holds.*

Therefore, we have shown that with overparameterization (required by Lemma 4), when $\mathbf{w}_r$ is relatively stable during training for all $r \in [m]$, i.e., the maximum change on $\mathbf{w}_r(t)$ is upper bounded during training (characterized by the $\ell_2$-norm of weight change restricted by $R$), $\mathbf{H}(t)$ is also relatively stable and remains positive definite with high probability.

## 4.2 CONVERGENCE OF THE IBP ROBUST LOSS

Next, we can derive the upper bound of the IBP loss during training. In the following lemma, we show that when $\mathbf{H}(t)$ remains positive definite, the IBP loss $\overline{L}(t)$ descends in a linear convergence rate, and meanwhile we have an upper bound on the change of $\mathbf{w}_r(t)$ w.r.t. time $t$:

**Lemma 6.** *Suppose for* $0 \leq s \leq t$, $\lambda_{\min}(\mathbf{H}(t)) \geq \frac{\lambda_0}{2}$, *we have*

$$\overline{L}(t) \leq \exp\left(2\overline{L}(0)\right)\overline{L}(0)\exp\left(-\frac{\lambda_0 t}{2}\right), \quad \|\mathbf{w}_r(t) - \mathbf{w}_r(0)\|_2 \leq \frac{nt}{\sqrt{m}}.$$

This lemma is proved in Appendix A.5, which follows the proof of Lemma 5.4 in Zou et al. (2018). To guarantee that $\lambda_{\min}(\mathbf{H}(s)) \geq \frac{\lambda_0}{2}$ for $0 \leq s \leq t$, by Lemma 5, we only require $\frac{nt}{\sqrt{m}} \leq R = \frac{c\delta\lambda_0}{d^{1.5}n^2}$, which holds sufficiently by

$$t \leq \frac{c\delta\lambda_0\sqrt{m}}{d^{1.5}n^3}. \tag{9}$$

Meanwhile, for each example $i$, the model can be certified by IBP on example $i$ with any $\ell_\infty$ perturbation within radius $\epsilon$, if and only if $\underline{u}_i > 0$, and this condition is equivalent to $l(\underline{u}_i) < \kappa$, where $\kappa := \log(1 + \exp(0))$. Therefore, to reach zero training error on the whole training set at time $t$, we can require $\overline{L}(t) < \kappa$, which implies that $\forall 1 \leq i \leq n, l(\underline{u}_i) < \kappa$. Then with Lemma 6, we want the upper bound of $\overline{L}(t)$ to be less than $\kappa$:

$$\overline{L}(t) \leq \exp\left(2\overline{L}(0)\right)\overline{L}(0)\exp\left(-\frac{\lambda_0 t}{2}\right) < \kappa,$$

which holds sufficiently by

$$t > \frac{4}{\lambda_0}\left(\log\left(\frac{\overline{L}(0)}{\kappa}\right) + \overline{L}(0)\right). \tag{10}$$

To make Eq. (10) reachable at some $t$, with the constraint in Eq. (9) we require:

$$\frac{4}{\lambda_0}\left(\log\left(\frac{\overline{L}(0)}{\kappa}\right) + \overline{L}(0)\right) < \frac{c\delta\lambda_0\sqrt{m}}{d^{1.5}n^3}. \tag{11}$$

The left-hand-side of Eq. (11) can be upper bounded by

$$\frac{4}{\lambda_0}\left(\log\left(\frac{\overline{L}(0)}{\kappa}\right) + \overline{L}(0)\right) = \frac{4}{\lambda_0}(\overline{L}(0) + \log(\overline{L}(0)) - \log(\kappa)) \le \frac{4}{\lambda_0}(2\overline{L}(0) - \log(\kappa)).$$

Therefore, in order to have Eq. (11) hold, it suffices to have

$$\frac{4}{\lambda_0}(2\overline{L}(0) - \log(\kappa)) < \frac{c\delta\lambda_0\sqrt{m}}{d^{1.5}n^3} \implies \overline{L}(0) + c_0 < \frac{c'\delta\lambda_0^2\sqrt{m}}{d^{1.5}n^3}, \tag{12}$$

where $c' := \frac{c}{8}$ and $c_0$ are positive constants.

Since $\overline{L}(0)$ has randomness from the randomly initialized weight $\mathbf{W}$, we need to upper bound the value of $\overline{L}(0)$ as we show in the following lemma (proved in Appendix A.6 by concentration):

**Lemma 7.** *In natural training, for any confidence $\delta(0 < \delta < 1)$, with probability at least $1 - \delta$, $L(0) = O(\frac{n}{\delta})$ holds. In IBP training, for any confidence $\delta(0 < \delta < 1)$, with probability at least $1 - \delta$, $\overline{L}(0) = O(\frac{n\sqrt{m}d\epsilon}{\delta} + \frac{n}{\delta})$ holds.*

And this lemma implies that with large $n$ and $m$, there exist constants $c_1, c_2, c_3$ such that

$$\overline{L}(0) \le \frac{c_1 n\sqrt{m}d\epsilon}{\delta} + \frac{c_2 n}{\delta} + c_3. \tag{13}$$

Plug Eq. (13) into Eq. (12), then the requirement in Eq. (12) can be relaxed into:

$$\frac{c'\delta\lambda_0^2\sqrt{m}}{d^{1.5}n^3} > \frac{c_1 n\sqrt{m}d\epsilon}{\delta} + \frac{c_2 n}{\delta} + c_3 + c_0 \implies \left(\frac{c'\delta\lambda_0^2}{d^{1.5}n^3} - \frac{c_1 nd\epsilon}{\delta}\right)\sqrt{m} > \frac{c_2 n}{\delta} + c_4, \tag{14}$$

where $c_4 := c_3 + c_0$ is a constant. As long as Eq. (14) holds, Eq. (11) also holds, and thereby IBP training is guaranteed to converge to zero IBP robust error on the training set.

## 4.3 PROVING THE MAIN THEOREM

Finally, we are ready to prove the main theorem. To make Eq. (11) satisfied, we want to make its relaxed version, Eq. (14) hold by sufficiently enlarging $m$. This requires that the coefficient of $\sqrt{m}$ in Eq. (14), $\frac{c'\delta\lambda_0^2}{d^{1.5}n^3} - \frac{c_1 nd\epsilon}{\delta}$ to be positive, and we also plug in the constraint on $\epsilon$ in Lemma 1:

$$\frac{c'\delta\lambda_0^2}{d^{1.5}n^3} - \frac{c_1 nd\epsilon}{\delta} > 0, \quad \epsilon \le \frac{\sqrt{2d}R}{\log(\sqrt{\frac{2\pi d}{R}}\xi)}.$$

Combining these two constraints, we can obtain the constraint for $\epsilon$ in the main theorem:

$$\epsilon < \min\left(\frac{c'\delta^2\lambda_0^2}{c_1 d^{2.5}n^3}, \frac{\sqrt{2d}R}{\log(\sqrt{\frac{2\pi d}{R}}\xi)}\right).$$

Then by Eq. (14), our requirement on width $m$ is

$$m \ge \Omega\left(\left(\frac{d^{1.5}n^4\delta\lambda_0}{\delta^2\lambda_0^2 - \epsilon d^{2.5}n^4}\right)^2\right).$$

This completes the proof of the main theorem.s In our analysis, we focus on IBP training with $\epsilon > 0$. But IBP with $\epsilon = 0$ can also be viewed as standard training. By setting $\epsilon = 0$, if $m \ge \Omega(\frac{n^8 d^3}{\lambda_0^4\delta^4})$, our result implies that for any confidence $\delta$ $(0 < \delta < 1)$, standard training with logistic loss also converges to zero training error with probability at least $1 - \delta$. And as $\epsilon$ gets larger, the required $m$ for convergence also becomes larger.

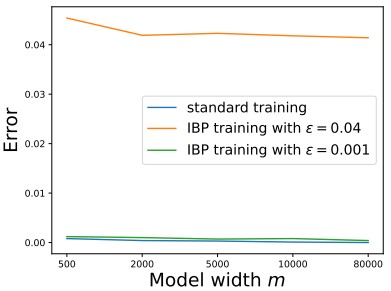
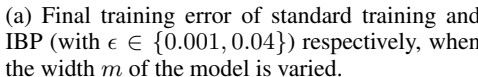
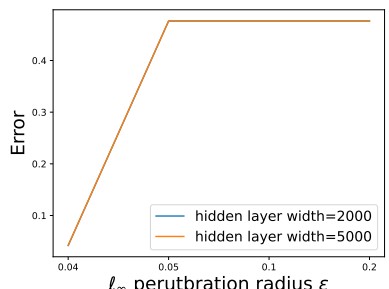

(a) Final training error of standard training and IBP (with $\epsilon \in \{0.001, 0.04\}$) respectively, when the width $m$ of the model is varied.

(b) Final training error of IBP training (on models with width 2000 and 5000 respectively), when the perturbation radius $\epsilon$ is varied.

Figure 1: Experimental results.

## 5 EXPERIMENTS

We further conduct experiments to compare the convergence of networks with different widths $m$ for natural training and IBP training respectively. We use the MNIST (LeCun et al., 2010) dataset and take digit images with label 2 and 5 for binary classification. And we use a two-layer fully-connected ReLU network with a variable width. We train the model for 70 epochs with SGD, and we keep $\epsilon$ fixed throughout the whole training process. We present results in Figure 1. First, compared with standard training, for the same width $m$, IBP has higher training errors (Figure 1a). Second, for relatively large $\epsilon$ ($\epsilon = 0.04$), even if we enlarge $m$ up to 80,000 limited by the memory of a single GeForce RTX 2080 GPU, IBP error is still far away from 0 (Figure 1a). This is consistent with our main theorem that when $\epsilon$ is too large, simply enlarging $m$ cannot guarantee the convergence. Moreover, when $\epsilon$ is even larger, IBP training falls into a local minimum of random guess (with errors close to 50%) (Figure 1b). We conjecture that this is partly because $\lambda_0$ can be very small with a large perturbation, and then the training can be much more difficult, and this difficulty cannot be alleviated by simply enlarging the network width $m$. Existing works with IBP-based training typically use a scheduling on $\epsilon$ and gradually increase $\epsilon$ from 0 until the target value for more stable training. Overall, the empirical observations match our theoretical results.

## 6 CONCLUSION

In this paper, we present the first theoretical analysis of IBP-based certified robust training, and we show that IBP training can converge to zero training error with high probability, under certain conditions on perturbation radius and network width. Meanwhile, since the IBP robust accuracy is a lower bound of the true robust accuracy (see Section 3.2), upon convergence the true robust accuracy also converges to 100% on training data and the certification by IBP accurately reflects the true robustness. Our results have a condition requiring a small upper bound on $\epsilon$, and it will be interesting for future work to study how to relax this condition, take the effect of $\epsilon$ scheduling into consideration, and extend the analysis to deeper networks.

## ACKNOWLEDGEMENTS

We thank the anonymous reviewers for their helpful comments. This work is partially supported by NSF under IIS-2008173, IIS-2048280 and by Army Research Laboratory under agreement number W911NF-20-2-0158; QG is partially supported by the National Science Foundation CAREER Award 1906169 and IIS-2008981. The views and conclusions contained in this paper are those of the authors and should not be interpreted as representing any funding agencies.

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

# A  PROOF OF LEMMAS

## A.1  PROOF OF LEMMA 1

*Proof.* For all $i \in [n], r \in [m]$, we first consider the change of indicator $\mathbb{1}(\mathbf{w}_r(t)^\top \mathbf{x}_i \pm \epsilon \|\mathbf{w}_r(t)\|_1 > 0)$ during training compared to the value at $t = 0$ (the notation $\pm$ here means the analysis is consistent for both $+$ and $-$ cases). Under the constraint that $\|\mathbf{w}_r(t) \pm \mathbf{w}_r(0)\|_2 \le R$ and $\|\mathbf{x}_i\|_\infty \in [0, 1]^d$, we have (see Appendix B.2 for details):

$$\left| \mathbf{w}_r(t)^\top \mathbf{x}_i \pm \epsilon \|\mathbf{w}_r(t)\|_1 - (\mathbf{w}_r(0)^\top \mathbf{x}_i \pm \epsilon \|\mathbf{w}_r(0)\|_1) \right| \le (1 + \epsilon)\sqrt{d}R. \tag{15}$$

Thereby, if $\mathrm{sign}(\mathbf{w}_r(t)^\top \mathbf{x}_i \pm \epsilon \|\mathbf{w}_r(t)\|_1) \ne \mathrm{sign}(\mathbf{w}_r(0)^\top \mathbf{x}_i \pm \epsilon \|\mathbf{w}_r(0)\|_1)$, then at initialization, we must have

$$|\mathbf{w}_r(0)^\top \mathbf{x}_i \pm \epsilon \|\mathbf{w}_r(0)\|_1| \le (1 + \epsilon)\sqrt{d}R. \tag{16}$$

We want to upper bound the probability that Eq. (16) holds. It is easy to show that if the following two inequalities hold, then Eq. (16) does not hold for sure:

$$|\mathbf{w}_r(0)^\top \mathbf{x}_i| \ge 2(1 + \epsilon)\sqrt{d}R, \tag{17}$$

$$\epsilon \|\mathbf{w}_r(0)\|_1 \le (1 + \epsilon)\sqrt{d}R. \tag{18}$$

Therefore,

$$\Pr\left( |\mathbf{w}_r(0)^\top \mathbf{x}_i \pm \epsilon \|\mathbf{w}_r(0)\|_1| \le (1 + \epsilon)\sqrt{d}R \right)$$

$$\le 1 - \Pr\left( |\mathbf{w}_r(0)^\top \mathbf{x}_i| \ge 2(1 + \epsilon)\sqrt{d}R \text{ and } \|\mathbf{w}_r(0)\|_1 \le (1 + \epsilon)\sqrt{d}R \right).$$

For Eq. (17), by anti-concentration inequality for Gaussian, we have

$$\Pr(|\mathbf{w}_r(0)^\top \mathbf{x}_i| \le 2(1 + \epsilon)\sqrt{d}R) \le \frac{4(1 + \epsilon)\sqrt{d}R}{\sqrt{2\pi}\xi}. \tag{19}$$

In other words, with probability at least $1 - 4(1 + \epsilon)\sqrt{d}R/(\sqrt{2\pi}\xi)$, Eq. (17) holds. And for Eq. (18), by the tail bound of standard Gaussian and union bound, we have

$$\Pr(\epsilon \|\mathbf{w}_r(0)\|_1 \le (1 + \epsilon)\sqrt{d}R) \ge 1 - 2d \exp\left( -\frac{2(1 + \epsilon)^2 dR^2}{\epsilon^2} \right). \tag{20}$$

Combining Eq. (19) and Eq. (20), Eq. (16) holds with at most the following probability

$$\frac{4(1 + \epsilon)\sqrt{d}R}{\sqrt{2\pi}\xi} + 2d \exp\left( -\frac{2(1 + \epsilon)^2 dR^2}{\epsilon^2} \right). \tag{21}$$

Here we require $\epsilon$ to be sufficiently small such that

$$\frac{(1 + \epsilon)\sqrt{d}R}{\sqrt{2\pi}\xi} \ge d \exp\left( -\frac{2(1 + \epsilon)^2 dR^2}{\epsilon^2} \right) \tag{22}$$

and we can solve the inequality to obtain an upper bound for $\epsilon$ (detailed in Appendix B.3):

$$\epsilon \le \frac{\sqrt{2d}R}{\log(\sqrt{\frac{2\pi d}{R}}\xi)}, \tag{23}$$

and in this case Eq. (21) holds with probability at least

$$\frac{6}{\sqrt{2\pi}\xi}(1 + \epsilon)\sqrt{d}R.$$

Therefore, we upper bound the probability:

$$\Pr\Big(\text{sign}(\mathbf{w}_r(t)^\top \mathbf{x}_i \pm \epsilon \|\mathbf{w}_r(t)\|_1) \neq \text{sign}(\mathbf{w}_r(0)^\top \mathbf{x}_i \pm \epsilon \|\mathbf{w}_r(0)\|_1)\Big) \leq \frac{6}{\sqrt{2\pi}\xi}(1+\epsilon)\sqrt{d}R.$$

Thereby

$$\forall i \in [n], r \in [m], \;\; \Pr(A_{ri}^+(t) \neq A_{ri}^+(0)), \Pr(A_{ri}^-(t) \neq A_{ri}^-(0)) \leq \frac{6}{\sqrt{2\pi}\xi}(1+\epsilon)\sqrt{d}R.$$

Note that at least one of $A_{ri}^+(t)$ and $A_{ri}^-(t)$ always remains zero during training, because condition $y_i a_r = 1$ in $A_{ri}^+(t)$ and condition $y_i a_r = -1$ in $A_{ri}^-(t)$ are mutually exclusive. Then

$$\Pr(A_{ri}^+(t) + A_{ri}^-(t) \neq A_{ri}^+(0) + A_{ri}^-(0)) \leq \frac{6}{\sqrt{2\pi}\xi}(1+\epsilon)\sqrt{d}R,$$

$$\Pr(A_{ri}^+(t) - A_{ri}^-(t) \neq A_{ri}^+(0) - A_{ri}^-(0)) \leq \frac{6}{\sqrt{2\pi}\xi}(1+\epsilon)\sqrt{d}R.$$

Next we can upper bound the probability that each of $\alpha_{rij}(t), \beta_{rij}(t), \gamma_{rij}(t)$ $(\forall i, j \in [n], r \in [m])$ changes respectively:

$$\Pr(\alpha_{rij}(t) \neq \alpha_{rij}(0)), \Pr(\beta_{rij}(t) \neq \beta_{rij}(0)), \Pr(\gamma_{rij}(t) \neq \gamma_{rij}(0)) \leq \frac{12}{\sqrt{2\pi}\xi}(1+\epsilon)\sqrt{d}R.$$

$\square$

### A.2 PROOF OF LEMMA 2

*Proof.* With Lemma 1, we can bound the expectation of the change for each element in $\mathbf{H}(t)$ (Eq. (6)) as:

$$\mathbb{E}[|\mathbf{H}_{ij}(t) - \mathbf{H}_{ij}(0)|]$$
$$\leq \frac{1}{m}\mathbb{E}\Big(m\tilde{R}\|\mathbf{x}_i\|_2\|\mathbf{x}_j\|_2 + \epsilon m\tilde{R}\big((\|\mathbf{x}_i\|_2 + \|\mathbf{x}_j\|_2)\|\,\text{sign}(\mathbf{w}_r(t))\|_2\big) + \epsilon^2 dm\tilde{R}\Big)$$
$$\leq \tilde{R}d(1 + 2\epsilon + \epsilon^2)$$
$$= \frac{12(1+\epsilon)(1+2\epsilon+\epsilon^2)d^{1.5}}{\sqrt{2\pi}} \quad (\forall i, j \in [n])$$

Then by Markov's inequality, we have that with probability at least $1 - \delta$,

$$\|\mathbf{H}(t) - \mathbf{H}(0)\|_2 \leq \sum_{i \in [n], j \in [n]} |\mathbf{H}_{ij}(t) - \mathbf{H}_{ij}(0)| \leq \frac{12(1+\epsilon)(1+2\epsilon+\epsilon^2)d^{1.5}n^2}{\sqrt{2\pi}\xi\delta}R.$$

$\square$

### A.3 PROOF OF LEMMA 3

*Proof.* First for simplicity, we define $\rho_i = -A_{ri}^+(0) + A_{ri}^-(0)$ $(\rho_i \in \{-1, 0, 1\})$, and

$$\phi(\mathbf{x}_i)(\mathbf{w}_r(0)) = y_i\Big(\mathbf{x}_i + \epsilon\rho_i \,\text{sign}(\mathbf{w}_r(0))\Big)\mathbb{1}\Big(\mathbf{w}_r(0)^\top\Big(\mathbf{x}_i + \epsilon\rho_i \,\text{sign}(\mathbf{w}_r(0))\Big) > 0\Big).$$

To prove that $\lambda_0 > 0$, similar as Theorem 3.1 in Du et al. (2019b), we need to prove that for any $r \in [m]$, if $\eta_1, \eta_2, ..., \eta_n$ $(\forall i \in [n], \eta_i \in \mathbb{R})$ satisfy $\sum_{i=1}^n \eta_i\phi(\mathbf{x}_i)(\mathbf{w}_r(0)) = 0$ almost everywhere (a.e.) for any $\mathbf{w}_r(0)$, we have $\forall i \in [n], \eta_i = 0$.

In Theorem 3.1 in Du et al. (2019b), it is proved that for $\phi'(\mathbf{x}_i)(\mathbf{w}) = \mathbf{x}_i\mathbb{1}(\mathbf{w}^\top \mathbf{x}_i)$ $(i \in [n])$, when $\forall i \neq j, \mathbf{x}_i \nparallel \mathbf{x}_j$ holds, for any $\eta_1, \eta_2, ..., \eta_n(\forall i \in [n], \eta_i \in \mathbb{R})$, if

$$\sum_{i=1}^n \eta_i\phi'(\mathbf{x}_i)(\mathbf{w}_r(0)) = 0,$$

then $\forall i \in [n], \eta_i = 0$. For any $r \in [m]$, by taking $\forall i \in [n], \mathbf{x}'_i = \mathbf{x}_i + \epsilon\rho_i \operatorname{sign}(\mathbf{w}_r(0))$, we have $\phi(\mathbf{x}_i)(\mathbf{w}_r(0)) = \phi'(\mathbf{x}'_i)(\mathbf{w}_r(0))$, and it holds that $\mathbf{x}'_i \in B_\infty(\mathbf{x}_i, \epsilon), \mathbf{x}'_j \in B_\infty(\mathbf{x}_j, \epsilon)$. Then if $\mathbf{x}_i \nparallel \mathbf{x}_j, \forall i, j, \eta_1, \eta_2, ..., \eta_n$ satisfy $\sum_{i=1}^n \eta_i \phi'(\mathbf{x}'_i)(\mathbf{w}_r(0)) = 0$ a.e., we have $\forall i \in [n], \eta_i = 0$.

Therefore if $\forall i, j \in [n], i \neq j, \forall \mathbf{x}'_i \in B_\infty(\mathbf{x}_i, \epsilon), \forall \mathbf{x}'_j \in B_\infty(\mathbf{x}_j, \epsilon), \mathbf{x}'_i \nparallel \mathbf{x}'_j$, if $\eta_1, ..., \eta_n$ satisfy

$$\sum_i \eta_i \phi(\mathbf{x}_i)(\mathbf{w}_r(0)) = 0,$$

then

$$\sum_i \eta_i \phi'(\mathbf{x}'_i)(\mathbf{w}_r(0)) = 0$$

also holds, and then $\forall i \in [n], \eta_i = 0$. $\qquad\square$

### A.4 PROOF OF LEMMA 5

*Proof.* The lemma can be proved by solving inequality

$$\lambda_{\min}(\mathbf{H}(t)) \geq \lambda_{\min}(\mathbf{H}(0)) - \frac{12(1+\epsilon)(1+2\epsilon+\epsilon^2)d^{1.5}n^2}{\sqrt{2\pi}\xi\delta}R \geq \frac{\lambda_0}{2}. \tag{24}$$

According to Lemma 4, $\lambda_{\min}(\mathbf{H}(0)) \geq \frac{3}{4}\lambda_0$. And with Eq. (8), in order to ensure $\lambda_{\min}(\mathbf{H}(t)) \geq \frac{\lambda_0}{2}$, we can make

$$\frac{12(1+\epsilon)(1+2\epsilon+\epsilon^2)d^{1.5}n^2}{\sqrt{2\pi}\xi\delta}R \leq \frac{\lambda_0}{4}.$$

This yields

$$R \leq \frac{\sqrt{2\pi}\xi\delta\lambda_0}{48(1+\epsilon)(1+2\epsilon+\epsilon^2)d^{1.5}n^2}.$$

Note that $0 \leq \epsilon \leq 1$, and thus $1 + \epsilon \leq 2$ and $1 + 2\epsilon + \epsilon^2 \leq 4$ can be upper bounded by constants respectively. Then we can take

$$R \leq \frac{\sqrt{2\pi}\xi\delta\lambda_0}{384d^{1.5}n^2} = \frac{c\delta\lambda_0}{d^{1.5}n^2}, \quad \text{where } c = \frac{\sqrt{2\pi}\xi}{384},$$

and in this case $\lambda_{\min}(\mathbf{H}(t)) \geq \frac{\lambda_0}{2}$ w.p. at least $1 - \delta$ probability. $\qquad\square$

### A.5 PROOF OF LEMMA 6

*Proof.* The proof of this lemma is inspired by the proof of Lemma 5.4 in Zou et al. (2018). In our proof, we define $f(\mathbf{x}) = (f(x_1), f(x_2), ..., f(x_n))$, where $f(x)$ is a scalar function and $\mathbf{x}$ is a vector of length $n$. When $\lambda_{\min}(\mathbf{H})(s) \geq \frac{\lambda_0}{2}$ holds for $0 \leq s \leq t$, we can bound the derivative of $\overline{L}(t)$:

$$\frac{d\overline{L}(\mathbf{u})}{dt} = \sum_{i=1}^{n} l'(\underline{u}_i)\frac{\partial \underline{u}_i}{\partial t}$$

$$= -\sum_{i=1}^{n} l'(\underline{u}_i)\sum_{j=1}^{n} l'(\underline{u}_j)\mathbf{H}_{ij}$$

$$= -l'(\mathbf{u})^{\top}\mathbf{H}l'(\mathbf{u})$$

$$\overset{(i)}{\leq} -\frac{\lambda_0}{2}\sum_{i=1}^{n} l'(u_i)^2$$

$$\overset{(ii)}{\leq} \frac{\lambda_0}{2}\sum_{i=1}^{n} l'(u_i)$$

$$\overset{(iii)}{\leq} -\frac{\lambda_0}{2}\sum_{i=1}^{n} \min(1/2, \frac{l(u_i)}{2})$$

$$\overset{(iv)}{\leq} -\frac{\lambda_0}{2}\min\left(1/2, \sum_{i=1}^{n}\frac{l(u_i)}{2}\right)$$

$$= -\frac{\lambda_0}{2}\min(1/2, \frac{\overline{L}(\mathbf{u})}{2})$$

$$\overset{(v)}{\leq} -\frac{\lambda_0}{2}\frac{1}{2 + 2/\overline{L}(\mathbf{u})},$$

where (i) is due to $\lambda_{\min}(\underline{\mathbf{H}}) \geq \lambda_0$, (ii) is due to $-l'(u) \leq 1$, (iii) holds due to the following property of cross entropy loss $-l'(u) \geq \min(1/2, \frac{l(u)}{2})$, (iv) holds due to the function $\min(1/2, x)$ is a concave function and Jenson's inequality, (v) holds due to $\min(a,b) \geq \frac{1}{1/a+1/b}$

Therefore, we have

$$2\frac{d\overline{L}(\mathbf{u})}{dt} + \frac{2}{\overline{L}(\mathbf{u})}\frac{d\overline{L}(\mathbf{u})}{dt} \leq -\frac{\lambda_0}{2}.$$

By integration on both sides from $0$ to $t$, we have

$$\overline{L}(\mathbf{u}(t)) - \overline{L}(\mathbf{u}(0)) + \log\left(\overline{L}(\mathbf{u}(t))\right) - \log\left(\overline{L}(\mathbf{u}(0))\right) \leq -\frac{\lambda_0 t}{4}.$$

Therefore, we have

$$\log\left(\overline{L}(\mathbf{u}(t))\right) \leq -\frac{\lambda_0 t}{4} + \overline{L}(\mathbf{u}(0)) + \log\left(\overline{L}(\mathbf{u}(0))\right),$$

which yields

$$\overline{L}(\mathbf{u}(t)) \leq \exp\left(-\frac{\lambda_0 t}{4}\right)\exp\left(\overline{L}(\mathbf{u}(0))\right)\overline{L}(\mathbf{u}(0)).$$

And we can bound the change of $\mathbf{w}_r$.

$$\left\|\frac{d\mathbf{w}_r(t)}{dt}\right\|_2 = \left\|\frac{d\overline{L}(t)}{d\mathbf{w}_r}\right\|_2$$

$$= \left\|\sum_{i=1}^{n} l'(\underline{u}_i)\frac{1}{\sqrt{m}}a_r y_i \sigma'(\langle\mathbf{w}_r, \mathbf{x}_i \pm \epsilon\|\mathbf{w}_r\|_1\rangle)(\mathbf{x}_i \pm \epsilon\|\mathbf{w}_r\|_1)\right\|_2$$

$$\leq \frac{1}{\sqrt{m}}\sum_{i=1}^{n}\|l'(\underline{u}_i)\|_2$$

$$\leq \frac{n}{\sqrt{m}},$$

where $\sigma'(\cdot)$ stands for the derivative of the ReLU activation. Thus

$$\|\mathbf{w}_r(t) - \mathbf{w}_r(0)\|_2 \leq \frac{nt}{\sqrt{m}}.$$

$\square$

## A.6 PROOF OF LEMMA 7

*Proof.* We first prove the standard training part. As we have defined previously that

$$L(0) = \sum_{i=1}^n \log(1 + \exp(-u_i(0))),$$

where

$$u_i(0) = y_i \frac{1}{\sqrt{m}} \sum_{r=1}^m a_r \sigma(\mathbf{w}_r(0)^\top \mathbf{x}_i).$$

For each $a_r \sigma(\mathbf{w}_r(0)^\top \mathbf{x}_i), r \in [m], i \in [n]$, note that the randomness only comes from random initialization for $\mathbf{w}_r$, there is $\frac{1}{2}$ possibility that it is equal to 0, and another $\frac{1}{2}$ possibility that it follows a normal distribution $\mathcal{N}(0, \sigma_i^2)$, where $\sigma_i = \|\mathbf{x}_i\|_2^2$. Therefore, we have

$$\mathbb{E}(a_r \sigma(\mathbf{w}_r(0)^\top \mathbf{x}_i)) = 0,$$

$$\mathrm{Var}(a_r \sigma(\mathbf{w}_r(0)^\top \mathbf{x}_i)) = \frac{\sigma_i^2}{2},$$

$$\mathbb{E}\left(\frac{1}{\sqrt{m}} \sum_{r=1}^m a_r \sigma(\mathbf{w}_r(0)^\top \mathbf{x}_i)\right) = 0,$$

$$\mathrm{Var}\left(\frac{1}{\sqrt{m}} \sum_{r=1}^m a_r \sigma(\mathbf{w}_r(0)^\top \mathbf{x}_i)\right) = \frac{\sigma_i^2}{2}.$$

Therefore, by Chebyshev's inequality, we can bound

$$\Pr(|u_i(0)| \leq \frac{\sigma_i^2}{2\delta}) \geq 1 - \delta.$$

And we can bound $L(0) = O(\frac{n \max_{i=1}^n \sigma_i^2}{2\delta}) = O(\frac{n}{\delta})$ with probability at least $1 - \delta$.

For IBP training,

$$\underline{u}_i(0) = \frac{1}{\sqrt{m}} \sum_{r=1}^m \mathbb{1}(y_i a_r = 1)\sigma\big(\mathbf{w}_r(0)^\top \mathbf{x}_i - \epsilon\|\mathbf{w}_r(0)\|_1\big) + \mathbb{1}(y_i a_r = -1)\sigma\big(\mathbf{w}_r(0)^\top \mathbf{x}_i + \epsilon\|\mathbf{w}_r(0)\|_1\big).$$

Thus

$$|\underline{u}_i(0) - u_i(0)| \leq \frac{1}{\sqrt{m}} m\epsilon\|\mathbf{w}_r(0)\|_1 = \sqrt{m}\epsilon\|\mathbf{w}_r(0)\|_1.$$

By $\mathbb{E}(\|\mathbf{w}_r\|_1) = O(d)$ and Markov's inequality, with probability at least $1 - \delta$,

$$|\underline{u}_i(0) - u_i(0)| \leq O(\frac{\sqrt{m}d\epsilon}{\delta}).$$

And we can bound $\overline{L}(0) = O(\frac{n\sqrt{m}d\epsilon}{\delta} + \frac{n}{\delta})$ with probability at least $1 - \delta$. $\square$

## B  DETAILED DERIVATION FOR OTHER EQUATIONS OR INEQUALITIES

### B.1  DERIVATION ON THE DYNAMICS OF $\underline{u}_i(t)$

We provide a detailed derivation on the dynamics of $\underline{u}_i(t)$ presented in Eq. (5), which we use $\mathbf{H}_i(t)$ to describe $\frac{d}{dt}\underline{u}_i(t)$:

$$
\begin{aligned}
\frac{d}{dt}\underline{u}_i(t) &= \sum_{r=1}^{m} \left\langle \frac{\partial \underline{u}_i(t)}{\partial \mathbf{w}_r(t)}, \frac{d\mathbf{w}_r(t)}{dt} \right\rangle \\
&= \sum_{r=1}^{m} \left\langle \frac{\partial \underline{u}_i(t)}{\partial \mathbf{w}_r(t)}, -\frac{\partial \overline{L}(\mathbf{W}(t), \mathbf{a})}{\partial \mathbf{w}_r(t)} \right\rangle \\
&= \sum_{r=1}^{m} \left\langle \frac{\partial \underline{u}_i(t)}{\partial \mathbf{w}_r(t)}, -\sum_{j=1}^{n} l'(\underline{u}_j) \frac{\partial \underline{u}_j(t)}{\partial \mathbf{w}_r(t)} \right\rangle \\
&= \sum_{j=1}^{n} -l'(\underline{u}_j) \sum_{r=1}^{m} \left\langle \frac{\partial \underline{u}_i(t)}{\partial \mathbf{w}_r(t)}, \frac{\partial \underline{u}_j(t)}{\partial \mathbf{w}_r(t)} \right\rangle \\
&= \sum_{j=1}^{n} -l'(\underline{u}_j) \mathbf{H}_{ij}(t),
\end{aligned}
$$

### B.2  DERIVATION FOR EQ. (15)

Eq. (15) basically comes by triangle inequality:

$$
\begin{aligned}
&\left| \mathbf{w}_r(t)^\top \mathbf{x}_i - \epsilon \|\mathbf{w}_r(t)\|_1 - (\mathbf{w}_r(0)^\top \mathbf{x}_i - \epsilon \|\mathbf{w}_r(0)\|_1) \right| \\
&= \left| (\mathbf{w}_r(t) - \mathbf{w}_r(0))^\top \mathbf{x}_i - \epsilon \|\mathbf{w}_r(t)\|_1 + \epsilon \|\mathbf{w}_r(0)\|_1 \right| \\
&\leq \left| (\mathbf{w}_r(t) - \mathbf{w}_r(0))^\top \mathbf{x}_i \right| + \epsilon \left| \|\mathbf{w}_r(t)\|_1 - \|\mathbf{w}_r(0)\|_1 \right| \\
&\leq \left| (\mathbf{w}_r(t) - \mathbf{w}_r(0))^\top \mathbf{x}_i \right| + \epsilon \left\| \mathbf{w}_r(t) - \mathbf{w}_r(0) \right\|_1 \\
&\leq (1 + \epsilon)\sqrt{d}R.
\end{aligned}
$$

### B.3  DERIVATION FOR EQ. (23)

We solve the inequality in Eq. (22) to derive an upper bound for $\epsilon$ in Eq. (23):

$$
\frac{1}{\sqrt{2\pi}\xi}(1+\epsilon)\sqrt{d}R \geq \frac{1}{\sqrt{2\pi}\xi}\sqrt{d}R \geq d\exp\left(-\frac{2(1+\epsilon)^2 dR^2}{\epsilon^2}\right),
$$

$$
\frac{R}{\sqrt{2\pi d}\xi} \geq \exp\left(-\frac{2(1+\epsilon)^2 dR^2}{\epsilon^2}\right),
$$

$$
\log\left(\frac{R}{\sqrt{2\pi d}\xi}\right) \geq -\frac{2(1+\epsilon)^2 dR^2}{\epsilon^2},
$$

and then we can require

$$
\log\left(\frac{R}{\sqrt{2\pi d}\xi}\right) \geq -\frac{2dR^2}{\epsilon^2} \geq -\frac{2(1+\epsilon)^2 dR^2}{\epsilon^2} \quad \Rightarrow \epsilon \leq \frac{\sqrt{2d}R}{\log\left(\sqrt{\frac{2\pi d}{R}}\xi\right)}.
$$

