# OpenReview forum: "On the Convergence of Certified Robust Training with Interval Bound Propagation"
_ICLR.cc/2022/Conference — ICLR 2022 Poster_

### Official Review · Reviewer_Ysed · 2021-11-01

**Correctness:** 3
**Technical Novelty And Significance:** 4
**Empirical Novelty And Significance:** Not applicable
**Recommendation:** 6
**Confidence:** 3

**Main Review:**

This paper explores the training dynamics of IBP training and provides a convergence analysis. This is a novel direction for certified robust training and as such valuable for the research community. Specifically, the authors restrict themselves to a (presumably simpler then the general) setting of 2 layer ReLU networks, where only the weights of the first layer will get changed during training, the weights of the second layer remain unchanged and are either -1 or 1.  The authors then proceed to establish various relationships with between the (time dependent) network weights, and eigenvalues of a matrix governing the training dynamics. Finally they establish that a growing network width will with high probability lead to a convergence of the IBP certified robust error is zero. The high level ideas seem to check out, the proofs where sporadically checked.

The background section (Section 3) provides a good introduction to the techniques the paper builds upon. However, Section 4 needs to be improved. Between the many Lemmas, the central theme is sometimes not clear. Section 4 would benefit from adding more explanations and more intuition. To give an example, the step from Eq. 30 to Eq. 31 seems not obvious. Further, if i understood the paper correctly, it should be clarified in the conclusion that the certified robust accuracy converges to 100% on the training set.

Further Questions for the authors:
- Please speculate: Do you expect a similar result for arbitrary depth instead of arbitrary width?
- Does Assumption 1 hold for the MNIST dataset? If, not, can this be fixed by offsetting all pixel values by a small positive constant?
- Why is Assumption 2 needed intuitively? Could this assumption potentially be relaxed or do you expect that then the theorem would not hold anymore?
- What are the the technical similarities and differences to Du et al. (2018b)?

Minor:
- Eq. 6: Here, it could be clarified notationally that $\bar{L}$ is obtained using IBP.
- Eq. 11: One “-“ to much?
- Eq. 12: Clarify that the ‘ in l’ denotes the derivative.
- Theorem 1: Do I suppose correctly the authors mean here that the IBP certified robust error converges on the training set to zero with high probability instead of the IBP certified robust accuracy?

**Summary Of The Paper:**

This work provides a theoretical analysis on the convergence of IBP training on overparametrized networks. The main Theorem states that the (IBP) certified robust training error can converge to zero with high probability.

**Summary Of The Review:**

This paper explores the certified robust training dynamics and proves convergence with high probability on a training set under certain assumptions. While the direction is novel, the writing and presentation should be improved.

---

> ### Author Response · Authors · 2021-11-18
> **Author response to Reviewer Ysed**
>
>
> We thank the reviewer for the review. We respond to some questions below:
>
> ## Arbitrary depth
>
> “Do you expect a similar result for arbitrary depth instead of arbitrary width?”
>
> For IBP training, large depth can lead to loose bounds in deeper layers, which can harm the trainability of IBP training, so IBP training is typically used with relatively shallow networks. Intuitively we think we need to balance the network capacity and bound tightness if we want to increase the depth of networks, and arbitrarily increasing the depth may not necessarily lead to better IBP training. Also, to analyze the convergence of IBP training, deeper networks can bring difficulty in theoretical analysis, and the tightness of the convergence bounds will be another factor.
>
> ## Assumption 1
>
> “Does Assumption 1 hold for the MNIST dataset?”
>
> We think it doesn’t hold directly (because there are originally many black pixels in the dataset), but it can be easily satisfied by linearly normalizing the dataset, just as the way mentioned by the reviewer. Therefore we can set Assumption 1 without the loss of generality.
>
> ## Assumption 2
>
> “Why is Assumption 2 needed intuitively? ”
>
> Assumption 2 is needed to assure that the dataset is separable with perturbations. In Du et al. (2018b), there is a similar assumption that for any $i \neq j, x_i \nparallel x_j$. Intuitively, when the assumption does not hold, it is possible that two parallel training examples $x_i, x_j$ with different labels can have output $f(x_i) = \eta f(x_j)$ for some constant $\eta$ (the considered network does not have a bias term), and then the predicted class by the model for these two examples from different classes is always the same. In our paper, Assumption 2 further considers the effect of perturbations.
>
> ## Comparison to Du et al., 2018b
>
> Similarities: We both reach conclusions on the convergence by proving the stability of the Gram matrix, and then using the property that the Gram matrix remains positive definite to obtain the convergence.
>
> Differences:
> * We have a different Gram matrix, since we compute the derivatives of the output by IBP computation, rather than the standard output. Thereby, our Gram matrix contains terms related to $\epsilon$ (Eq. (20)), and also indicator terms (Eq. (17), Eq. (18) and Eq. (19)) that are also related to $\epsilon$.
> * Due to the additional terms on $\epsilon$, it becomes more difficult to prove the stability of the indicator terms (Lemma 1). We cannot simply apply an anti-concentration as done by Du et al., 2018b. We need to split  the indicator into two parts (one standard part without $\epsilon$, and the other part contains $\epsilon$). For the second part with $\epsilon$, we used a different technique by the tail bound of Gaussian. Then we also need to combine probabilities from these two parts.
> * Due to the additional $\epsilon$ factor, the initial IBP loss is dependent on the width of the network $m$, which affects the final result of convergence.
> * We use a logistic loss for the classification task (IBP is typically for classification), while Du et al., 2018b used a quadratic loss for a regression task. Due to the difference in the loss function, the proof of bounding the loss and the change of weight $\mathbf{w}_r$ (Lemma 6) is much different.
>
> ## Minor points
>
> We thank the reviewer for finding out these points, and we have revised our paper accordingly. For Theorem 1, the reviewer is correct that there is a typo. We have also rephrased Theorem 1 and Section 4 to make them more clear.

---

### Official Review · Reviewer_YCZg · 2021-11-02

**Correctness:** 4
**Technical Novelty And Significance:** 4
**Empirical Novelty And Significance:** Not applicable
**Recommendation:** 8
**Confidence:** 2

**Main Review:**

The authors build on previous work on proving the convergence of gradient descent for "natural" training of over-parametrized networks. In particular, relying on similar assumptions and derivation to (Du et al. 2018b), the authors extend the proofs to the case of the upper bound to the robust loss as given by IBP. It is proved that IBP training converges to zero certified robust loss with high probabiliy. The choice of IBP is relevant, as it forms the basis of most state-of-the-art certified training algorithms.

While the assumptions seem to be quite restrictive (two-layer ReLU network of constrained width, and the upper bound on the perturbation radius), the community could start from this work to provide similar extensions as those provided for natural training by (Allen-Zhu et al. 2019).
Furthermore, as corollary of the convergence proof, it is implied that the IBP certified robust accuracy will converge to the true robust accuracy: this complies with the folkore observation in the community that methods trained with a given algorithm are more easily verify with the same certification method. Moreover, I found the observations in the experimental section to be of interest to the community. In particular, point (b), providing a possible explanation for the commonly employed epsilon warm-up schedules.

Minor comments:
- In Theorem 1 it is said "the IBP certified robust accuracy [...] can converge to zero": do the authors mean "the IBP certified robust loss"?
- epsilon=0.03 is actually a fairly small value, as opposed to epsilon=0.1 or epsilon=0.3, which are more commonly employed in MNIST.
- in the conclusions, "our results has a condition" -> "our results have a condition"


**Summary Of The Paper:**

The paper presents (to the best of my knowledge) the first convergence analysis of IBP training, a method commonly employed to train networks that are certifiably robust to adversarial examples.

**Summary Of The Review:**

The authors non-trivially extend theoretical analyses previously presented in the context of natural training of overparametrized networks to the context of IBP training. While the assumptions that lead to the results are fairly restrictive, I believe the results are of great interest to the community and could lay the ground for further work in the area.

---

> ### Author Response · Authors · 2021-11-18
> **Author response for Reviewer YCZg**
>
> We thank the reviewer for identifying the value of our work. For the reviewer’s comments:
>
> ## Typos
>
> For Theorem 1, the reviewer is correct that it is a typo. And we have rephrased Theorem 1 to make it more clear. We have also fixed the typo in the conclusion mentioned by the reviewer.
>
> ## About $\epsilon$
>
> Yes, in existing IBP-based training works, usually $\epsilon \in [0.1,0.2,0.3,0.4]$ is used for MNIST, but they need a scheduling on $\epsilon$ and gradually increase $\epsilon$ from 0 until the target value during the training, otherwise the training will have a poor performance or even diverge. We have not considered this scheduling in our theoretical analysis so far, and if we directly use $\epsilon=0.1$ without the scheduling, a two-layer MLP model cannot reach a reasonable certified robust accuracy (See figure 1b). It will be interesting to further analyze the effect of this $\epsilon$ scheduling in future work.

---

> > ### Comment · Reviewer_YCZg · 2021-11-29
> > **Update**
> >
> > I thank the authors for their response.
> >
> > After reading the other reviews and the relative authors' rebuttals, I confirm my rating of 8: in spite of the limitations pointed out by the other reviewers (restrictive assumptions), I believe this work is of great interest to the community, as it provides the first results on the convergence of IBP training.
> > I hope further work in the area will follow.

---

### Official Review · Reviewer_n3Ki · 2021-11-02

**Correctness:** 3
**Technical Novelty And Significance:** 2
**Empirical Novelty And Significance:** 1
**Recommendation:** 5
**Confidence:** 4

**Main Review:**

Although I am aware of the line of research “certifiably robust training of neural networks”, I have not seen any work on the analysis of convergence of the gradient descent algorithm, especially in the IBP setting. Hence, assuming there is really no work doing this, I believe this paper studies a very essential and relevant problem. The mathematical steps look correct to me.

My major concerns:

1 - **Presentation.** Overall, I believe the paper does not introduce the literature thoroughly, includes multiple typos, and has mistakes in the terminology. I will list some of those at the end of my review, but I think in general the paper should undergo a major review (language, terminology, editorial positioning, presentation of the existing literature, presentation of the main theorem, and the proofs).

2 - **Generality and Assumptions.** Sections 1-3 are on known work and Section 5 is not very critical since this paper proves a convergence result theoretically.  Hence, this paper’s contribution can be summarised by Theorem 1 (and the Lemmas leading to it). This result is for i) overparametrised networks, ii) box perturbations with small enough radius , iii) +-1 classification problems, iv) ReLU networks, v) single and wide hidden layer. Even in the presence of these assumptions, the results are “with high probability”. Although having some assumptions is essential (e.g., bounding the perturbation radius), most results follow from the specific set of assumptions that work for this case but cannot be easily extended (e.g., the classes being +-1 is helping largely to look at the sign of the linear expressions to classify, etc.). Moreover, the upper bound on the error radius decays if we require a stronger result, not the other way around. Moreover, currently, if we fix an architecture then we should decrease the error radius until the result works.

Minor points:
- Typos: “overparameteried” (abstract), “overparameterized” (page 2), “maxmization” (page 3), “Pr(|w_r(0)^T x_i )” (page 7)
- “Training can converge to a point” is being used frequently but not clear what it is
- The perturbations are additive, but this is not mentioned. There are other well-known perturbations including multiplicative, affine, or functional transformations.
- End of Section 1: “under certain conditions” -> this is not clear unless someone reads the whole paper. Could the author(s) please mention the conditions briefly?
- Section 2.1: “based on linear relaxation” -> linear relaxation “of what” is not clear
- Section 2.2: “to a globally optima” -> optima is plural
- problem(1): (x,y) \sim \mathcal{X} is not clarified as \mathcal{X} is a training set (please explain how an expectation is being taken over the training set)
- Page 3: “Inner maximization is achieved” is not very clear
- Page 3: “You et al. (2021) showed that adversarial training provably learn robust halfspaces in the presence of noise” -> the context of ‘robust half spaces’ is not clarified (it does not have meaning alone)
- Section 2.3: Last paragraph follows simply from the definition of robust optimisation: we typically minimise theta for the worst-case solution which depends on theta so the inner problem cannot be solved numerically rather its tractable counterpart is derived analytically. This is well-known already.
- Page 4: u is not defined when l(u) is being introduced
- Page 4 onward: sometimes u_i is being used, sometimes u_i(W,a,x_i)
- Some sentences claim a result for \Delta_i, but do not define i (or state “for all i= 1,…,n)
- Assumption 2: “j” is not defined
- Definition of \overline{L} in equation (6) is not in line with the standard definitions (maybe in the summation you can already include the domains of maximisation problems as max_{ \Delta _i } \{log(…) | ||\Delta_i||_{\infty}\leq \epsilon  \}?
- Usage of “we only care about” is perhaps a bit informal
- Page 5: “to denote the value at time t” -> “value of what” is not clarified
- Page 5: A_{ri}^{+} and A_{ri}^{-} are being used before being defined. Also, the definitions are in general not clear (using = instead of := or mentioning “let … be “)
- “Least eigenvalue” -> smallest/minimum eigenvalue?
- Theorem 1 and throughout the paper: $\xi$ is never defined
- Referring to “Section 2” in the statement of Theorem 1 is perhaps a little bit informal
- Theorem 1 is overall very hard to read. Could the author(s) please split it to multiple sentences and present it more clearly?
- Section 4.3: in the beginning ||w_r(t) - w_r(0)||_2 \leq R is being used without defining “r” or “t”.
- Overall, Theorem 1 and Lemma 2 state at least “1-\delta” without defining \delta
- Lemma 3 mentions the result holds when Assumption 2 is satisfied, but Assumption 1 is not mentioned in the other results relying on that.
- Page 8: “then we want to make” is unclear
- Lemma 7: The sentences are not complete
- Page 8: “Plug in Eq. (30)…” is this a typo?
- Section 4: Last paragraph -> when epsilon = 0 the result should be well-known. Could the author(s) please cite relevant papers’ and the convergence mentioned there to show whether the results found in this paper match these results?
- Numerical experiments are not very clear. The MNIST dataset is being used but the paper relies on binary classification problems. Overall I think the numerical experiments could be explained more clearly.
- Conclusion: “which converges to 100%” terminology is not clear to me
- The references are not consistent: e.g., “In ICLR, 2015” versus “In International Conference on Learning Representations, 2020a”.
- Appendix: “at most 1 minus the probability …” this is informal
- I can share a further list of minor editorial suggestions should the chairs decide to accept this work.


**Summary Of The Paper:**

Training deep neural networks in the presence of adversarial perturbations (in the input data) is a very active research topic. There are lots of works defining notions of robustness, proposing solution algorithms, and introducing algorithmic improvements. Several of the recent techniques involve or extend the interval bound propagation (IBP) technique. However, there is no work analysing the convergence of IBP even in its simplest setting. In this paper, the author(s) analyse the convergence of IBP in a simplified setting for the first time.

**Summary Of The Review:**

The paper studies a very relevant problem. There are no mistakes in the proofs as far as I can see. However, the main concerns are (i) the paper is hard to read and understand, (ii) the convergence result relies on many assumptions that cannot easily be extended to more general settings, (iii) the convergence is probabilistic and if the desired probability gets "finer" then the required radius of the infinity-ball will get smaller hence one needs to also increase the number of parameters of the network so the results do not work for a fixed architecture or a fixed error radius.

---

> ### Author Response · Authors · 2021-11-18
> **Author response to Reviewer n3Ki (part 3/3)**
>
> ## Minor points
>
> We thank the reviewer for the editorial suggestions and we have revised the paper accordingly. And in response to some of the minor points:
>
> * “Typo overparameterized”
>
> We think “overparameterized” is not a typo in American English and is commonly used:
> https://scholar.google.com/scholar?hl=en&as_sdt=0%2C5&q=%22overparameterized%22&btnG=
>
> * “$\xi$ is never defined”
>
> It’s defined in Assumption 1, which is the lower bound of $\| \mathbf{x}_i \|_2 $. We’ve made the definition more clear.
>
> *  “A_{ri}^{+} and A_{ri}^{-}...”
>
> They are defined in the “where” clause. We’ve changed to use “:=”.
>
> * “The perturbations are additive, but this is not mentioned.”
>
> It is mentioned in the beginning of Section 3.2 .
>
> *  “linear relaxation of what is not clear”
>
> We have clarified as “the linear relaxation for nonlinear activation functions”.
>
> * “Overall, Theorem 1 and Lemma 2 state at least “1-\delta” without defining \delta”
>
> We have added “for any confidence $\delta(0<\delta<1) $”.
>
> * “ (x,y) \sim \mathcal{X} is not clarified”
>
> $\mathcal{X}$ stands for a data distribution for training, and we have revised the paper to clarify it.
>
> * “Page 3: “Inner maximization is achieved” is not very clear.”
>
> We have changed the sentence to “Empirical adversarial training approximately solves the inner maximization by running adversarial attacks”.
>
> * “Section 2.3: Last paragraph follows simply from the definition of robust optimisation: we typically minimise theta for the worst-case solution which depends on theta so the inner problem cannot be solved numerically rather its tractable counterpart is derived analytically. This is well-known already.”
>
> Here compare empirical adversarial training and certified robust training methods, such that due to the difference, prior theoretical analysis for adversarial training is not applicable to certified robust training. The difference in the training methods may be known to some readers, but we aim to emphasize the difference in the corresponding theoretical analysis.
>
> * “Conclusion: “which converges to 100%” terminology is not clear to me”
>
> We have rephrased the sentence: “Meanwhile, since the certified robust accuracy by IBP is a lower bound of the true robust accuracy (see Section 3.2), and we have shown that it can converge to 100% on the training set, the true robust accuracy also converges to 100% on the training set.”
>
> * “the context of ‘robust half spaces’ is not clarified”
>
> Halfspace is a well-defined mathematical concept and can be used as a machine learning model ($x\rightarrow \operatorname{sign}(\mathbf{w}^\top x) $). A robust halfspace stands for a halfspace model robust to perturbations/noise. Details are available in the original paper.
>
> * “Assumption 1 is not mentioned in the other results relying on that.”
>
> It is a basic assumption, which can be achieved by normalizing the dataset. All the subsequent results depend on on this assumption. We have added a sentence to clarify it.
>
> * Theorem 1
>
> We’ve rewritten Theorem 1 to make it more clear and formal.
>
> * “Training can converge to a point” is being used frequently but not clear what it is
>
> We have changed the word “point” into “state”. It means when training converges, with the weights at that time, the model satisfies the property mentioned in the following “where” clause (“where the robustness certification by IBP accurately reflects the true robustness of the model”).
>
> * “Section 4: Last paragraph -> when epsilon = 0 the result should be well-known.”
>
> This result is similar to Du et al. 2018b's: $m = \Omega(\frac{n^6}{\lambda_0^4 \delta^2})$. Note that Du et al. 2018b used a quadratic loss for regression, which is different from the logistic loss for classification here (since IBP is for classification). Thus, the results are not exactly the same. Also, our result depends on $d$, due to a relaxation related to $\epsilon$ in Lemma 2. If $\epsilon=0$, the dependence on $d$ can be avoided.
>
> * “Page 8: “Plug in Eq. (30)…” is this a typo?”
>
> We mean to use the upper bound of  $ \overline{L}(0) $ in Eq. (30) to get Eq. (31).
>
> * “Numerical experiments are not very clear. ”
>
> To make the experiments consistent with our theoretical analysis, we changed from using all the 10 classes in MNIST into using 2 classes only (using digit 2 and 5 only). We have updated the figure (in the revised paper), and the results are consistent with the the 10-class setting. We have rephrased the section to make it more clear.
>
> * Page 8: “then we want to make” is unclear
>
> We have rephrased Section 4.3. We want to make Eq. (28) satisfied, for which we only need to make Eq. (31) hold by sufficiently enlarging $m$. And this requires 1) the coefficient of $\sqrt{m}$ is positive; 2) $m$ is sufficiently large to make the inequality Eq. (31) hold. This leads to bounds on $\epsilon$ and $m$.
>
> For all the other minor points, we have also revised the paper following your suggestions.

---

> ### Author Response · Authors · 2021-11-18
> **Author response to Reviewer n3Ki (part 2/3)**
>
> ### Summary
> In summary, we believe using these assumptions here are reasonable and we hope the reviewer could reconsider the score after reading our clarification.
>
> References:
>
> [1] Du, Simon, et al. "Gradient descent finds global minima of deep neural networks." International Conference on Machine Learning. PMLR, 2019.
>
> [2] Ji, Ziwei, and Matus Telgarsky. "Polylogarithmic width suffices for gradient descent to achieve arbitrarily small test error with shallow relu networks." International Conference on Learning Representations, 2020.
>
> [3] Satpathi, Siddhartha, and R. Srikant. "The Dynamics of Gradient Descent for Overparameterized Neural Networks." Learning for Dynamics and Control. PMLR, 2021.
>
> [4] Chen, Zixiang, et al. "How much over-parameterization is sufficient to learn deep relu networks?." International Conference on Learning Representations, 2021.

---

> ### Author Response · Authors · 2021-11-18
> **Author response to Reviewer n3Ki (part 1/3)**
>
> We thank the reviewer for the review. For concerns regarding the assumptions, we believe there is misunderstanding on theoretical research in machine learning, and we address the concerns in detail. For the presentation, we thank the reviewer for the detailed editorial suggestions, and we have revised our paper following the suggestions and uploaded the revision. We also respond to some questions in the minor points.
>
> ## Probabilistic results (with a high probability)
>
> We believe it is impossible to have a deterministic result because **neural networks are randomly initialized**. Due to the randomness of the initialization, there exists some poor initialization that cannot lead to a satisfactory convergence. For example, if all the weights happen to be initialized as 0, the gradients in the network are always 0 and thus the training will get stuck at the initialization. Such bad cases are inevitable but their probability is very small. Hence we can only expect to have probabilistic results.
>
> Moreover, **probabilistic results are quite typical in the theoretical analysis of machine learning algorithms**, not only in the deep learning area, but also in more fundamental theories of machine learning (such as PAC learnability).
>
> ## Assumptions
>
> First of all, as far as we know, there is no previous work on the convergence analysis for IBP training. Therefore, we **keep the setting simple to begin the analysis, so that we can focus on tackling key challenges introduced in IBP training**, such as handling perturbation terms in the derivations. Even in this setting, the convergence analysis for IBP training is already non-trivial.
> Moreover, we clarify that most of these assumptions are either commonly used in many other theoretical works for neural networks (with standard training), or practically used in IBP training:
>
> ### ReLU activation
>
> **IBP training itself is typically used with ReLU activation in existing works for certified training.** This is more suitable for IBP training in comparison to more complex activation functions, because when a ReLU neuron is inactive (given the bounds of its input), IBP bounds are exact in this region (Figure 5 in Shi et al, 2021). Therefore, it is reasonable to also use ReLU in our theoretical analysis.
>
> ### Box perturbations
>
> **IBP is also typically used for $\ell_\infty$ perturbations (i.e., box perturbations).** Since IBP computes interval bounds (just like boxes), IBP is naturally suitable for such perturbations. $\ell_\infty$ perturbation is also a widely adopted setting in empirical adversarial training. Despite its simplicity, it is a well-defined setting, and it is good for research that focuses on training methods (in contrast to works on robustness specifications). Therefore, we also use $\ell_\infty$ perturbations in our analysis.
>
> ### Overparameterization
>
> **Overparameterization is an important assumption in existing works on analyzing the convergence of neural network training.** Even on standard network training, this is an important assumption so far ([1, 2]). It is reasonable for us to adopt this assumption for analyzing IBP training.
>
> For the concern on the convergence result for a fixed architecture, our result is reasonable since **the width of a network restricts the capacity of the network**. Even for standard training, the network width needs to be increased, if we want the training to reach zero training error with a higher probability. Similarly, for IBP training, we have to increase the network width to reach zero training robust error, if we want a higher probability, or a larger $\epsilon$. If $\epsilon$ is larger, the robust learning problem is harder, and thus if we want a stronger result (with a higher probability) with a fixed architecture, $\epsilon$ needs to be smaller.
>
> ### Multi-class classification
> We believe that the analysis can be extended to multi-class classification with cross-entropy loss, but this will involve more complex derivations. We aim to focus on key challenges in the convergence analysis for IBP training, such as handling perturbation terms in the derivations, while the number of classes is relatively minor. Moreover, even on standard training, recent theoretical works on analyzing the convergence also typically use a binary classification setting ([2, 3, 4]) or a regression setting ([1]).
>
> To extend our work to a $K$-class setting, we think there will mainly be three modifications: 1) The second layer weights $\mathbf{a}$ should be of size $K \times r$ for $K$ output logits. 2) We need to use cross entropy loss instead of logistic loss and bound all the $K$ logits. 3) When we calculate the Gram matrix $\mathbf{H}$ as in Section 3.4, we need to calculate the derivatives for all the logits, and we also need to track the training dynamics of all the $K$ bounded logits. Then, we can similarly obtain the convergence result by replacing the robust logistic loss by a robust cross-entropy loss computed from all the $K$ logits.

---

> ### Author Response · Authors · 2021-11-29
> **We hope the reviewer can read our response and update the review**
>
> Dear reviewer,
>
> As we are approaching the end of the discussion period, we would like to summarize our response again: 1) For generality and assumptions, we have clarified that most of the assumptions are actually consistent with previous works on the theoretical analysis for neural networks/machine learning or consistent with practical settings of IBP training. Thus they are reasonable, and we also provide a sketch on extending our analysis to the multiclass setting; 2) For the presentation, we have carefully revised the paper, and we have fixed and/or responded to all the issues pointed out by the reviewer.
>
> We hope the reviewer can read our response and reconsider the rating. And please let us know if you have any additional questions.
>
> Thanks!
> Paper 4436 Authors

---

> > ### Comment · Reviewer_n3Ki · 2021-11-30
> > **The authors improved the writing of the paper, and discuss extensions accurately. However, I am not comfortable with Theorem 1.**
> >
> > I would like to thank the authors for their detailed response.
> >
> > I would also like to clarify a couple of misunderstandings in the previous iteration, which I am taking full responsibility for.
> >
> > First of all, I am of course not expecting to see any deterministic work due to obvious reasons. When I quoted and said “with high probability”, I meant that, in places where probably mathematically rigorous words should be used, we had phrases such as “training with high probability”. This misunderstanding is due to my poor placement of this sentence, especially since I said this right after stating that the assumptions are too restrictive. I can see that in the revision the authors improved the presentation of the paper significantly.
> >
> > Similarly, I think overall, my minor comments have been misunderstood. I will list a couple of such examples:
> > - A link is shared claiming “overparameterized” is used in American English. But this does not change the fact that previously there were typos such as “overparameteried”. I wanted to show a couple of example typos to the authors just to make sure they improve the writing of the paper. Because this paper took a significant amount of time to understand, not because of the mathematical steps (which are, as said before, very precise and ‘modern’), rather due to the writing of the paper, the appearance of the assumptions, delta showing up in the main theorem without being defined, etc.
> > - When I said “\xi is never defined”, I did not mean “\xi never appears”. It does appear in Assumption 1, however, without stating what \xi is, saying that we assume this is a lower bound on the norm was really hard to follow and understand.
> > - As the authors noted “Halfspace is a well-defined mathematical concept”, I think they think I am not aware of what a halfspace is. Me saying “the context of ‘robust halfspaces’ is not clarified” should not imply “I, as a reviewer, do not know what a halfspace is”. I just want to kindly bring this to the attention of the authors, that the paper is not self-contained, and sometimes more explanations are needed. This sentence is “Zou et al. (2021) showed that adversarial training provably learns robust halfspaces in the presence of noise.”
> >
> > More importantly, I am not sure if I can fully adopt and understand how Theorem 1 is useful. Similar to the other reviewers, I am really enthusiastic about the fact that this work provides the first convergence result on IBP training. However, I do not understand the practical meaning of imposing such an upper bound on the perturbation radius (which is given in such problems in any “robust” setting) by a function of delta and \xi, despite making the theory possible. It is clear that having a larger \xi or lower \epsilon is in line with the requirements of this Theorem, which are both a way of saying “epsilon should be really small with respect to the magnitude of the training data”. In that case, the result does not look very surprising. Maybe the positioning of the paper can be different so that \xi and \epsilon are fixed, and we try to find the network architecture guaranteeing convergence with high probability. Yet I don’t feel comfortable enough with adjusting the perturbation radius according to all the given parameters, and I feel like the results are a bit “chaotic” now, or at least as a reader I am not able to share the enthusiasm.
> >
> > I am increasing my initial review score to “marginally below acceptance threshold” as my minor concerns are all clarified. Again, I would like to sincerely thank the authors for all the work and detailed responses provided.

---

> > > ### Author Response · Authors · 2021-11-30
> > > **Follow-up response**
> > >
> > >
> > > We thank the reviewer for clarifying the comments and acknowledging our improvements for the presentation. The suggestions are helpful for us to make the paper more clear and easier to understand.
> > >
> > > For Theorem 1, we want to further clarify that our theorem contains two folds:
> > > *  Given an $\epsilon$, with this **fixed** $\epsilon$, as long as it satisfies the upper bound of $\epsilon$ we derive in Theorem 1, we can find an architecture with sufficiently large width $m$, for guaranteed convergence with high probability. And given the dataset, $\xi$ is a fixed constant. So this result matches the expectation by the reviewer “so that \xi and \epsilon are fixed, and we try to find the network architecture guaranteeing convergence with high probability”.
> > > * When $\epsilon$ is too large and does not satisfy the upper bound for $\epsilon$, our result implies that so far IBP training may not be guaranteed to converge even with large $m$, which is consistent with empirical experiments -- when $\epsilon$ is large, IBP training cannot converge to low training errors even with large $m$, which is different from standard training. This result is also consistent with the practice that previous IBP works rely on a scheduling for $\epsilon$ (which gradually increases $\epsilon$ from 0 to the target $\epsilon$ value) to stabilize IBP training (Gowal et al., 2018; Shi et al., 2021), otherwise IBP indeed cannot converge under larger $\epsilon$ in practice.
> > >
> > > We hope our further clarification can address the reviewer's concern on the implication of the theorem, and we will revise our paper to interpret our results more clearly.

---

### Decision · Program_Chairs · 2022-01-20

**Decision:**

Accept (Poster)

**Comment:**

Verifying robustness of neural networks is an important application in machine learning. The submission takes on this challenge via the interval bound propagation (IBP) framework and provides a theoretical analysis on the training procedure. They establish, in the large network with case, that the certification via IBP reflects the robustness of the neural network. Despite the tensions between the changing architecture and the required accuracy, the results are insightful. The AC recommends the authors to revise the paper, correcting the significant amounts of typos and improve the presentation for its final version.